# Regio- and enantioselective remote hydroarylation using a ligand-relay strategy

Yuli He [1], Jiawei Ma [1], Huayue Song[1], Yao Zhang[1], Yong Liang [1]✉, You Wang [1]✉ & Shaolin Zhu [1,2]✉

The design of a single complicated chiral ligand to well-promote each step of an asymmetric cascade reaction is sometimes a formidable challenge in transition metal catalysis. In this work, a highly regio- and enantioselective Ni-catalysed migratory hydroarylation relay process has been achieved with the combination of two simple ligands, one which accomplishes chain-walking and the other causing asymmetric arylation. This formal asymmetric C(sp$^3$)−H arylation provides direct access to a wide range of structurally diverse chiral 1,1-diarylalkanes, a structural unit found in a number of bioactive molecules. The value of this strategy was further demonstrated by the Ni-catalysed migratory asymmetric 1,3-arylboration.

[1] State Key Laboratory of Coordination Chemistry, Jiangsu Key Laboratory of Advanced Organic Materials, Chemistry and Biomedicine Innovation Center (ChemBIC), School of Chemistry and Chemical Engineering, Nanjing University, Nanjing 210093, China. [2] School of Chemistry and Chemical Engineering, Henan Normal University, Xinxiang 453007, China. ✉email: yongliang@nju.edu.cn; wangyou@nju.edu.cn; shaolinzhu@nju.edu.cn

The synergistic combination of olefin isomerization and cross-coupling, nickel-catalysed[1–3] migratory hydrofunctionalization[4–9] of olefins has emerged as a complementary and versatile platform that can realize a wide variety of remote C(sp³)−H functionalizations[10–26]. So far, much progress has been made in the NiH-catalysed asymmetric *ipso*-hydrofunctionalization of alkenes[27–50]. However, designing a chiral ligand for nickel that could promote both chain-walking and subsequent regio- and enantioselective reductive coupling at a remote position remains a formidable challenge (Fig. 1a).

Overcoming the limitation of single metal catalysis [ML], multimetallic catalysis [$M_A L_A$ & $M_B L_B$] provides a complementary strategy that has been well exploited. The alternative reaction using multiligands has remained largely underexplored. Pioneering work in this area has shown that multiligand based binary ligand complexes [$ML_A L_B$] could be formed and are more reactive catalysts[51–53]. A catalyst mixture [$ML_A$ & $ML_B$] formed by multiligands was also used by Buchwald et al.[54] to broaden the substrate scope of a system using a single catalyst [$ML_A$]. Instead of designing a complex ligand, another unique mode of employing multiligands is to undergo multiligand relay catalysis via dynamic ligand exchange in cases where a single ligand fails to selectively or efficiently promote all the steps of the transformation. In this attractive but largely underexplored pathway, each ligand is only required to promote partial steps of the catalytic cycle. This concept was preliminarily explored by White's group in the Pd-catalysed non-asymmetric allylic oxidation system while using stoichiometric amount of benzoquinone or DMSO as another ligand for functionalization step[55–57]. Very recently, Fu et al.[58] and Mauleón et al.[59] demonstrated that the exchange of dynamic mutiligands with metals could be used to sequentially promote different steps in Cu-catalysed relay reactions. At the same time, our laboratory[60] successfully introduced this concept to the nickel-catalysed asymmetric migratory hydrofunctionalization process, wherein the entire catalytical cycle could be subsequently promoted by an achiral chain-walking ligand and a structurally simple chiral asymmetric coupling ligand[27–44] (Fig. 1b). Pursuing this theme, we hypothesized that migratory asymmetric hydroarylation which consists of an achiral ligand (**L**) promoting chain-walking and a chiral ligand (**L***)

**Fig. 1 Design plan: Synergistic combination of a chain-walking ligand (L) and an asymmetric arylation ligand (L*) to access chiral 1,1-diarylalkanes.** **a** Stereochemistry in remote NiH catalysis is still a challenging issue. **b** Multiligand-relay catalysis as a solution for asymmetric remote hydrofunctionalization. **c** Ni-catalysed multiligand-relay catalysis to access chiral 1,1-diarylalkanes.

promoting asymmetric arylation at benzylic position[61–65] could be possible, leading to the facile synthesis of an enantioenriched 1,1-diarylalkane, a biologically active pharmacophore (Fig. 1c).

In this work, we describe a highly regio- and enantioselective Ni-catalysed migratory hydroarylation relay process enabled by a multiligand relay catalysis strategy. By synergistic combination of a simple ligand for chain-walking and a known ligand for asymmetric arylation, a wide variety of enantioenriched 1,1-diarylalkanes can be rapidly obtained under mild conditions.

## Results and discussion

**Reaction design and optimization.** Our initial investigation focused on the enantioselective remote hydroarylation of 4-phenyl-1-butene (**1a**) with 4-iodoanisole (**2a**) (Fig. 2). It was found that $NiCl_2 \cdot glyme$ (glyme = ethylene glycol dimethyl ether) and the synergistic combination of a chain-walking ligand (**L**, 2,9-dimethyl-1,10-phenanthroline) and an asymmetric arylation

ligand (**L\***, (4 R,4′R)-1,1′-bis(3-(tert-butyl)-phenyl)-4,4′-di(heptan-4-yl)-4,4′,5,5′-tetrahydro-1H,1′H-2,2′-biimidazole)[31] with DMMS (dimethoxymethylsilane) could afford the desired migratory product 1-methoxy-4-(1-phenylbutyl)-benzene (**3a**) as a single regioisomer in 84% isolated yield and with 95% enantiomeric excess (ee) (entry 1). Control experiments revealed that these two ligands are both essential for simultaneous control of the regio- and stereochemistry, and poor regioselectivity was observed in the absence of the chain-walking ligand (entries 2–4). Importantly, lowering the loading of the chain-walking ligand to 0.4 mol% had little impact on the overall performance (entry 5). Increasing the chain-walking ligand loading and decreasing the arylation ligand loading led however to a moderate decrease of both the yield and the enantioselectivity (entry 6). An alternative chain-walking ligand (**L1**) was found to be competent but less effective than **L** (entry 7). Evaluation of arylation ligands revealed that **L\*** provided the highest ee (entry 1 vs. entries 8–10). Diminished yields were

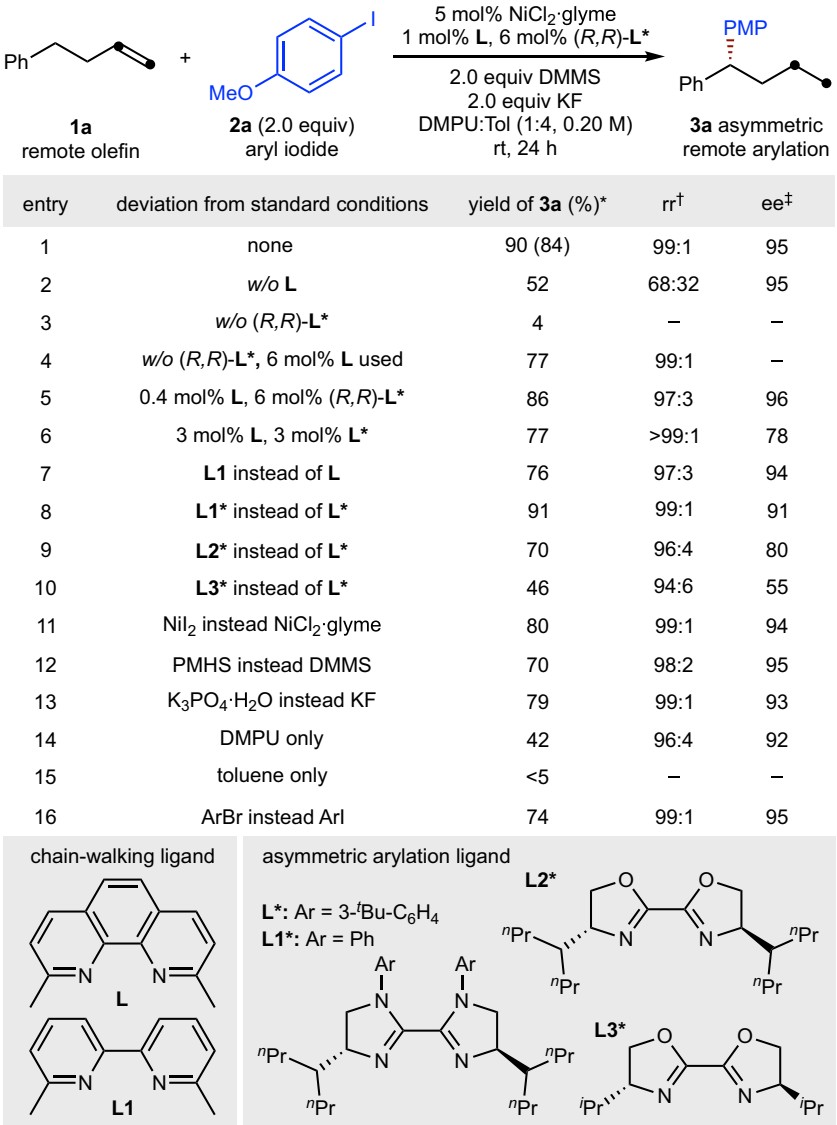

| entry | deviation from standard conditions | yield of **3a** (%)* | rr[†] | ee[‡] |
|---|---|---|---|---|
| 1 | none | 90 (84) | 99:1 | 95 |
| 2 | *w/o* **L** | 52 | 68:32 | 95 |
| 3 | *w/o* (R,R)-**L\*** | 4 | – | – |
| 4 | *w/o* (R,R)-**L\***, 6 mol% **L** used | 77 | 99:1 | – |
| 5 | 0.4 mol% **L**, 6 mol% (R,R)-**L\*** | 86 | 97:3 | 96 |
| 6 | 3 mol% **L**, 3 mol% **L\*** | 77 | >99:1 | 78 |
| 7 | **L1** instead of **L** | 76 | 97:3 | 94 |
| 8 | **L1\*** instead of **L\*** | 91 | 99:1 | 91 |
| 9 | **L2\*** instead of **L\*** | 70 | 96:4 | 80 |
| 10 | **L3\*** instead of **L\*** | 46 | 94:6 | 55 |
| 11 | $NiI_2$ instead $NiCl_2 \cdot$glyme | 80 | 99:1 | 94 |
| 12 | PMHS instead DMMS | 70 | 98:2 | 95 |
| 13 | $K_3PO_4 \cdot H_2O$ instead KF | 79 | 99:1 | 93 |
| 14 | DMPU only | 42 | 96:4 | 92 |
| 15 | toluene only | <5 | – | – |
| 16 | ArBr instead ArI | 74 | 99:1 | 95 |

**Fig. 2 Variation of reaction parameters.** *Yields were determined by gas chromatography (GC) analysis using *n*-dodecane as the internal standard, the yield within parentheses is the isolated yield and is an average of two runs (0.20 mmol scale). [†]Regioisomeric ratio (rr) represents the ratio of the major (1,1-diarylalkane) product to the sum of all other isomers as determined by GC and GC-MS analysis. [‡]Enantioselectivities were determined by chiral HPLC analysis, the absolute configuration of **4t** was determined by X-ray crystallography, and the configurations of the remaining products were assigned by analogy. Glyme ethylene glycol dimethyl ether, DMMS dimethoxymethylsilane, DMPU N,N′-dimethylpropyleneurea, Tol toluene, PMP p-methoxyphenyl, PMHS polymethylhydrosiloxane.

obtained when using other nickel sources (entry 11) or employing other silanes (entry 12) or replacing $K_3PO_4 \cdot H_2O$ as base (entry 13). The use of a single solvent led to either diminished yield (entry 14) or almost complete failure of the reaction (entry 15). Notably, a slightly reduced yield was obtained when a less reactive aryl bromide was used (entry 16).

**Substrate scope**. With the well-established conditions in hand, the scope and generality of the reaction were evaluated. As illustrated in Fig. 3, a wide variety of aryl and heteroaryl iodides or bromides are tolerated. In general, the less reactive aryl bromides resulted in a slightly decreased yield (3a, 3b, 3f, 3h, 3k, and 3t). The reaction proceeded well with both electron-rich (3b–3e) and electron-withdrawing aryl halides (3f–3q). A variety of functional groups are readily accommodated, including ethers (3a, 3e–3g, 3i, and 3s), esters (3b, 3k), a Boc carbamate (3c), an amide (3d), a trifluoromethyl group (3j), a nitrile (3l), an aryl fluoride (3m), and an aryl chloride (3n). Notably, under these exceptionally mild reaction conditions, sensitive functional groups such as aryl triflate (3o) commonly used for subsequent complementary cross-coupling, and readily reduced aldehydes (3p) and ketones (3q) were all unaffected. Heterocycles such as indole (3r) and pyridine (3s and 3t) are also competent coupling partners. However, o-substituted (hetero)aryl halides gave lower yields under current conditions.

We next explored the scope of alkenes (Fig. 4). As shown in Fig. 4a, a wide range of terminal aliphatic alkenes bearing a remote aryl (4b–4n, 4s, 4t) or heteroaryl (4o–4r) group undergo asymmetric migratory hydroarylation smoothly, regardless of the chain length between the C=C bond and the remote aryl group. In general, a slightly increased loading of the chain-walking ligand is beneficial for alkene substrates with a long chain (4m). A variety of substituents on the remote aromatic ring, including both electron-donating (4c–4e) and electron-withdrawing (4f–4l) substituents, are all well-tolerated. Notably, the 1,1-disubstituted alkene (4n) is also a suitable substrate, although the migratory product was obtained in decreased yield. As shown in Fig. 4b, unactivated internal alkenes are also suitable substrates (4u–4a'). Both E (4y, 4z) and Z (4a') alkenes, as well as E/Z mixtures (4u–4x) were suitable substrates. In addition, a range of different substituents at the other terminus of the alkyl chain, even with a heteroatomic substituent (4x–4z), were all well-tolerated, and arylation at benzylic position was still preferred.

**Application**. Benefiting from the chain-walking catalysis, isomeric mixtures of olefins could be used directly to produce the enantioenriched product (4m) in a regioconvergent fashion (Fig. 5a). The current multiple ligand catalysis could also be applied to migratory difunctionalization of alkenes[66–70]. As shown in Fig. 5b, a nickel-catalysed 1,3-arylboration reaction was

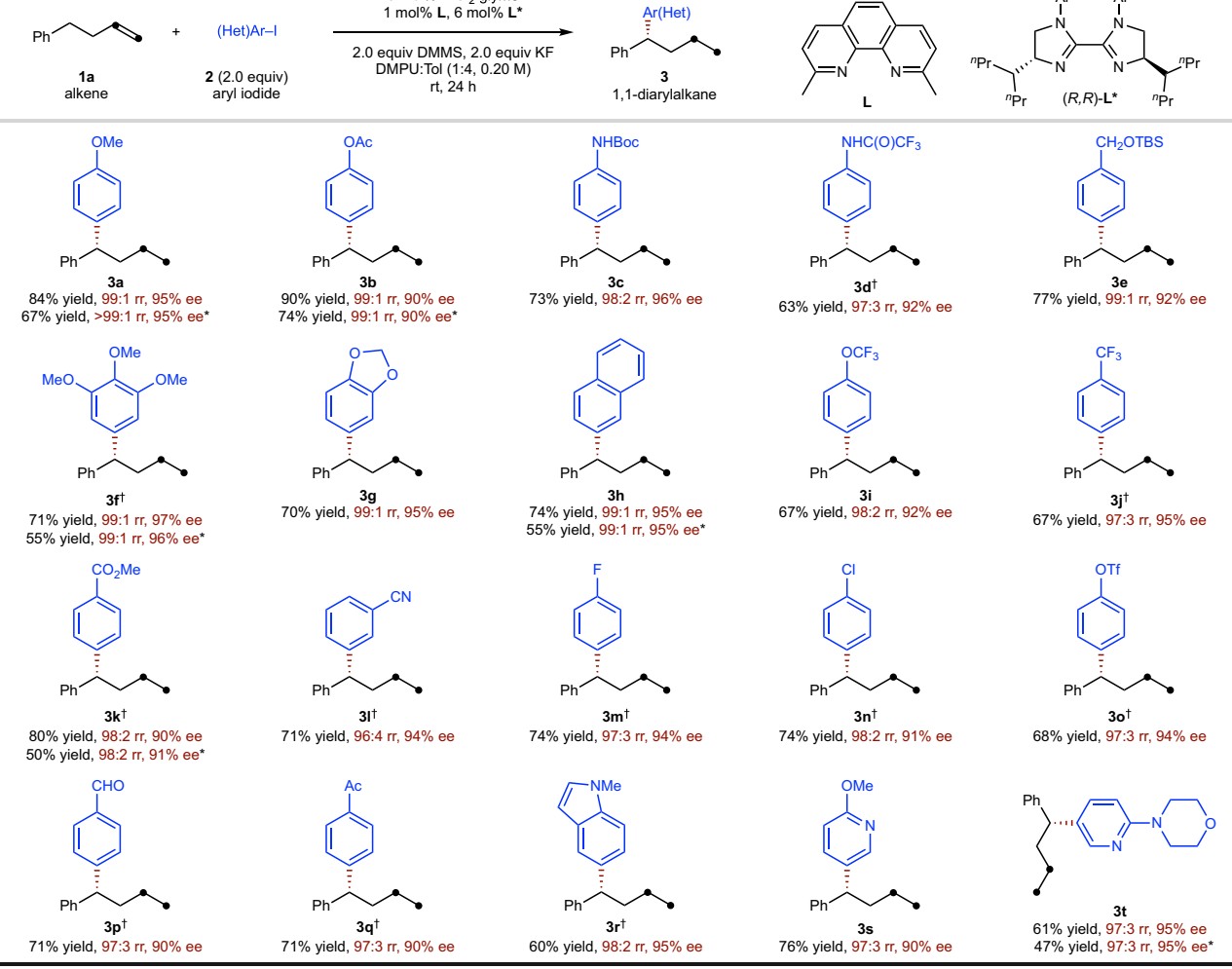

**Fig. 3 Substrate scope of aryl halide coupling partner.** Yield under each product refers to the isolated yield of purified product (0.20 mmol scale, average of two runs), regioisomeric ratio (rr) represents the ratio of the major (1,1-diarylalkane) product to the sum of all other isomers as determined by GC and GC-MS analysis, enantioselectivities were determined by chiral HPLC analysis. *Aryl bromide used. †0.4% **L** used.

carried out using a racemic ligand (**L2**)[66] to promote NiBpin insertion/chain-walking together with the asymmetric arylation ligand (**L***) which promotes asymmetric arylation. The desired migratory chiral products (**7a–7h**) were obtained with high regioselectivity and excellent ee.

**Mechanistic investigation.** A series of mechanistic experiments were carried out to understand the relay process. As shown in Fig. 6a, isomeric mixtures of olefins could be produced during the reaction process, an observation that is consistent with a fast chain-walking step. As shown in Fig. 6b, in the absence of cross-coupling partner, control experiment revealed that the alkene isomerization could also proceed smoothly with dual ligands or with a single chain-walking ligand. These results indicate that chain-walking precedes arylation without the participation of cross-coupling partner. In contrast, only a very small quantity of isomerized alkenes was observed while using arylation ligand **L*** alone. This observation is consistent with the conclusion that the chain-walking process is mainly promoted by achiral chain-walking ligand **L**. To probe whether hydronickellation is the enantio-determining step, isotopic labelling experiments of a cyclic styrene (**1b'**) substrate were

carried out with deuteropinacolborane (Fig. 6c). If the migratory insertion of NiD into styrene is the enantio-determining step, a diastereomerically pure product (**4b'-D**) should be obtained. However, a diastereomeric mixture was observed with asymmetric arylation ligand (**L***) alone or with dual ligand (**L/L***), and the diastereomeric ratio decreased while increasing the amount of achiral chain-walking ligand **L**. This observation indicates that the hydronickellation is not likely to be the enantio-determining step.

To better understand the reaction mechanism, we conducted a DFT study of the reaction pathways shown in Fig. 7 (see Supplementary Data 1). Previous computational studies have clarified that achiral Ni(II)H controls chain-walking process and gives the desired internal alkylnickel(II)[20,60], which undergoes transmetallation with Ni(I)H to generate the corresponding alkylnickel(I)[60]. Here we started from achiral alkylnickel(I) **INT1-a** and investigated the asymmetric arylation process. The direct oxidative addition of **INT1-a** with aryl iodide via **TS1-a** requires an activation free energy of 20.4 kcal/mol. Alternatively, **INT1-a** can first exchange its achiral bipyridine ligand with chiral bisimidazoline ligand to form either **INT1-R** or **INT1-S**. Although this process is endergonic by about 6 kcal/mol, the subsequent Ni(I)/Ni(III) oxidative

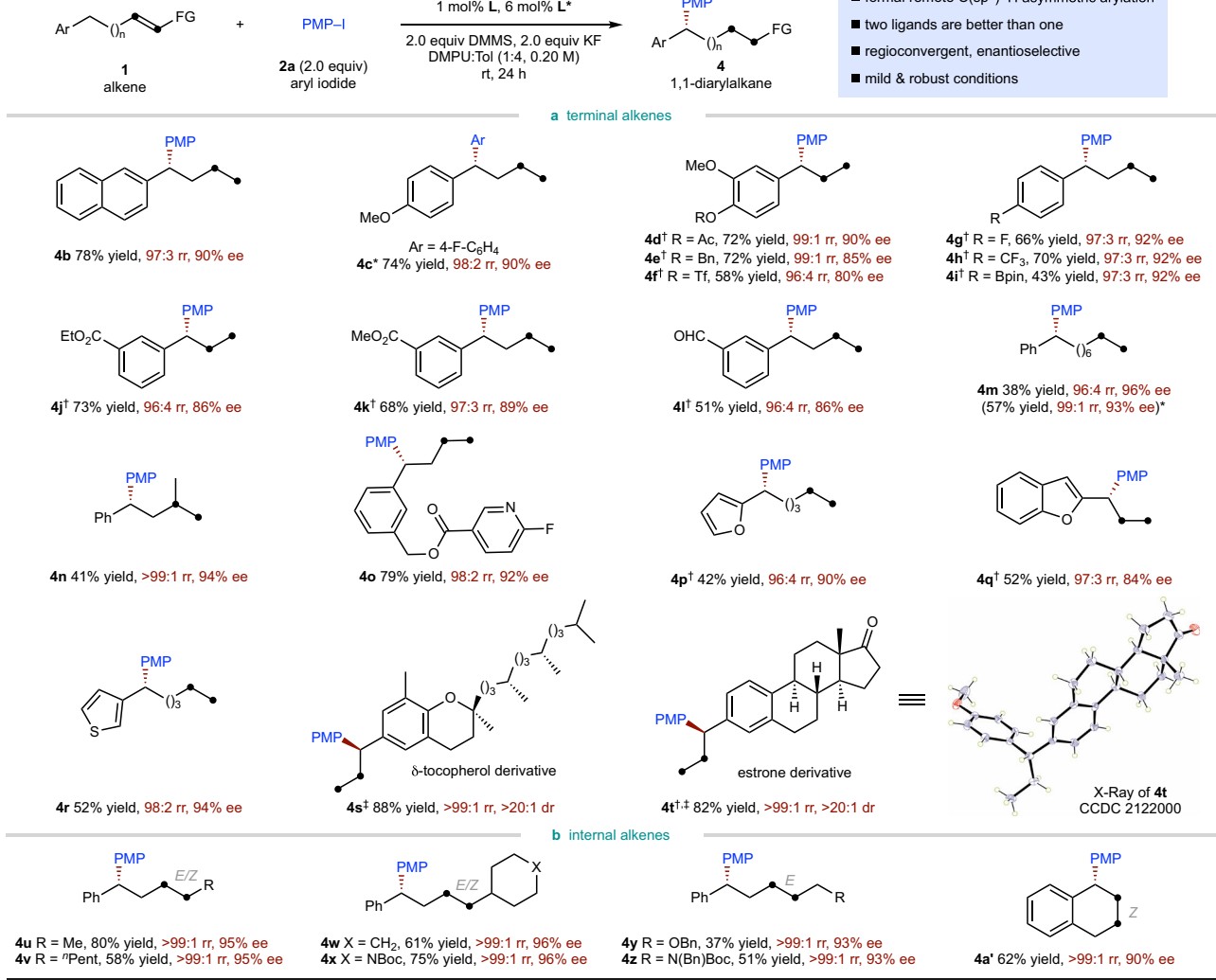

**Fig. 4 Substrate scope of alkene coupling component.** Under each product is given yield in percent, regioisomeric ratio (rr), and either the enantioselectivities (ee) or the diastereomeric ratio (dr). Yield, rr and ee are as defined in Fig. 3 legend. *2% **L** used. †0.4% **L** used. ‡Diastereoselectivity (dr) determined by [1]H NMR analysis of the crude reaction mixture.

**a** Regioconvergent, enantioselective, and bench-top set up experiment

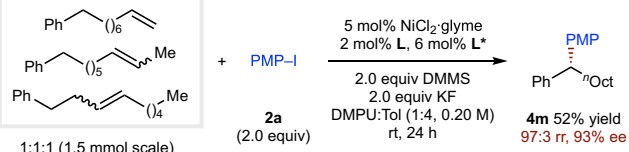

**b** Application of multiligand-relay catalysis (MLRC) in 1,3-arylboration reaction

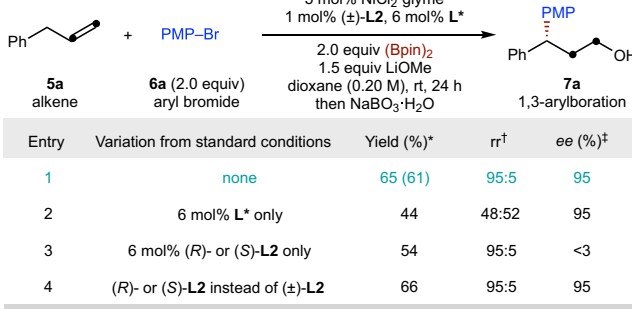

| Entry | Variation from standard conditions | Yield (%)* | rr† | ee (%)‡ |
|---|---|---|---|---|
| 1 | none | 65 (61) | 95:5 | 95 |
| 2 | 6 mol% L* only | 44 | 48:52 | 95 |
| 3 | 6 mol% (R)- or (S)-L2 only | 54 | 95:5 | <3 |
| 4 | (R)- or (S)-L2 instead of (±)-L2 | 66 | 95:5 | 95 |

*Yields determined by GC using n-dodecane as the internal standard, the yield in parentheses is the isolated yield. †rr determined by GC and GC-MS analysis. ‡Enantioselectivities determined by chiral HPLC analysis.

(±)-L2

selected examples

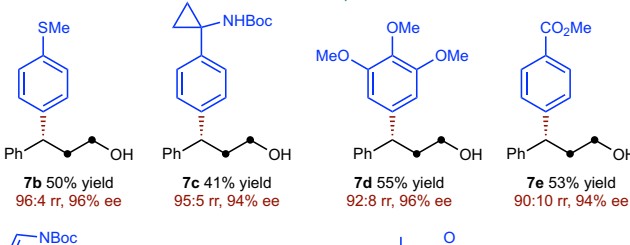

**7b** 50% yield
96:4 rr, 96% ee

**7c** 41% yield
95:5 rr, 94% ee

**7d** 55% yield
92:8 rr, 96% ee

**7e** 53% yield
90:10 rr, 94% ee

**7f** 66% yield
95:5 rr, 95% ee

**7g** 47% yield
91:9 rr, 93% ee

**7h** 44% yield
92:8 rr, 94% ee

**Fig. 5 Regioconvergent experiment and further application.**
**a** Regioconvergent, enantioselective, and bench-top set up experiment.
**b** Application of multiligand-relay catalysis in 1,3-arylboration reaction.

**a** Tracing experiment

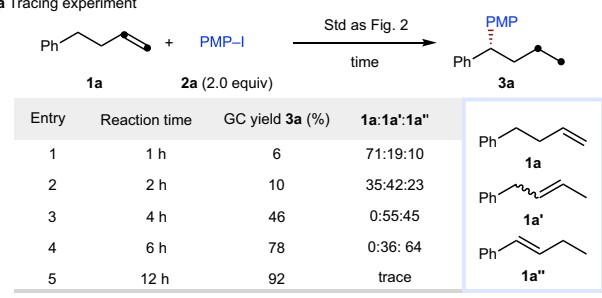

| Entry | Reaction time | GC yield 3a (%) | 1a:1a':1a'' |
|---|---|---|---|
| 1 | 1 h | 6 | 71:19:10 |
| 2 | 2 h | 10 | 35:42:23 |
| 3 | 4 h | 46 | 0:55:45 |
| 4 | 6 h | 78 | 0:36: 64 |
| 5 | 12 h | 92 | trace |

**b** Olefin isomerization in the absense of aryl iodide

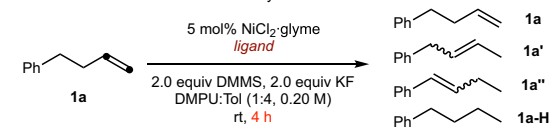

| isomer | | 1 mol% L | 6 mol% L* | 1 mol% L + 6 mol% L* |
|---|---|---|---|---|
| Ph | 1a | 12% | 83% | 60% |
| Ph | 1a' | 34% | 3% | 9% |
| Ph | 1a'' | 23% | 7% | 14% |
| Ph | 1a-H | 31% | 7% | 17% |

**c** NiD experiment: NiD syn-hydrometalation is not the enantio-determining step

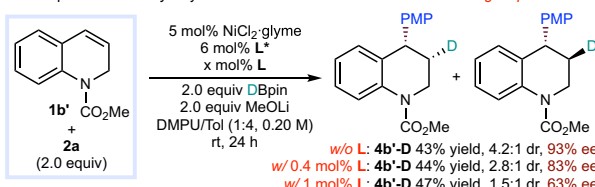

w/o L: **4b'-D** 43% yield, 4.2:1 dr, 93% ee
w/ 0.4 mol% L: **4b'-D** 44% yield, 2.8:1 dr, 83% ee
w/ 1 mol% L: **4b'-D** 47% yield, 1.5:1 dr, 63% ee

**Fig. 6 Tracing experiment, olefin isomerization experiment, and isotopic labelling experiment. a** Tracing experiment. **b** Olefin isomerization in the absence of aryl iodide. **c** NiD experiment: NiD syn-hydrometalation is not the enantio-determining step.

additions involving chiral ligand via **TS1-R** and **TS1-S** are much easier. The overall barrier for the formation of chiral Ni(III) species **INT2-R** is 16.2 kcal/mol via **TS1-R**, which is 4.2 kcal/mol lower than that for the oxidative addition of achiral alkylnickel(I) via **TS1-a** (20.4 kcal/mol). For two competing enantioselective transition states **TS1-R** and **TS1-S**, there is a 1.8 kcal/mol preference toward (R)-intermediate, corresponding to a 91% ee value of the final product at 298 K. In **TS1-R**, a less sterically demanding phenyl is placed close to the bulky isopropyl group of ligand, while in **TS1-S**, it is the ethyl group that generates steric repulsions with the bulky isopropyl group (highlighted in pink, Fig. 7). Meanwhile, the unreacted **INT1-S** can be rapidly converted to **INT1-R** via a reversible ligand exchange, transmetallation and alkene insertion process (see Supplementary Fig. 5). After the irreversible formation of chiral Ni(III) intermediate **INT2-R**, the following reductive elimination via **TS2-R** to yield product is very facile with a barrier of only 4.1 kcal/mol, which is much lower than the possible homolysis via **TS3-R**[62]. Therefore, the chiral bisimidazoline ligand controlled the Ni(I)/Ni(III) oxidative addition with aryl iodide, which is the rate-

and enantioselectivity-determining step in the asymmetric arylation.

In conclusion, by the combination of a chain-walking ligand and an asymmetric arylation ligand used in nickel chemistry, we have developed a highly enantioselective remote hydroarylation protocol. This mild, general and attractive route allows rapid access to an array of enantioenriched 1,1-diarylalkanes. Further application of this multiligand relay catalysis to simplify the chiral ligand design in nickel and other metal-catalysed multistep asymmetric reactions are currently in progress.

## Methods

**General procedure (A) for the regio- and enantioselective C(sp³)−H arylation.** In a nitrogen-filled glove box, to an oven-dried 8 mL screw-cap vial equipped with a magnetic stir bar was added NiCl₂·glyme (2.2 mg, 5.0 mol%), KF (23.2 mg, 2.0 equiv), L* (7.2 mg, 6.0 mol%), L (0.42 mg, 0.2 mL stock solution, 2.1 mg/mL in toluene), anhydrous toluene (0.60 mL) and DMPU (0.20 mL). The mixture was stirred for 15 min at rt, at which time 4-phenyl-1-butene (30 μL, 0.20 mmol), 4-iodoanisole (94.0 mg, 0.40 mmol) and DMMS (49.3 μL, 0.40 mmol) were added to the resulting mixture in this order. The tube was sealed with a teflon-lined screw cap, removed from the glove box and the reaction was stirred at rt (22 ~ 26 °C) for up to 24 h (the mixture was stirred at 750 rpm, ensuring that the base was uniformly suspended). After the reaction was complete, the reaction mixture was directly filtered through a short pad of silica gel (using EtOAc in petroleum ether) to give the crude product. n-Dodecane (20 μL) was added as an internal standard for GC analysis. The product was purified by chromatography on silica gel for each substrate. The yields reported are the average of at least two experiments, unless otherwise indicated. The enantiomeric excesses (% ee) were determined by HPLC analysis using chiral stationary phases.

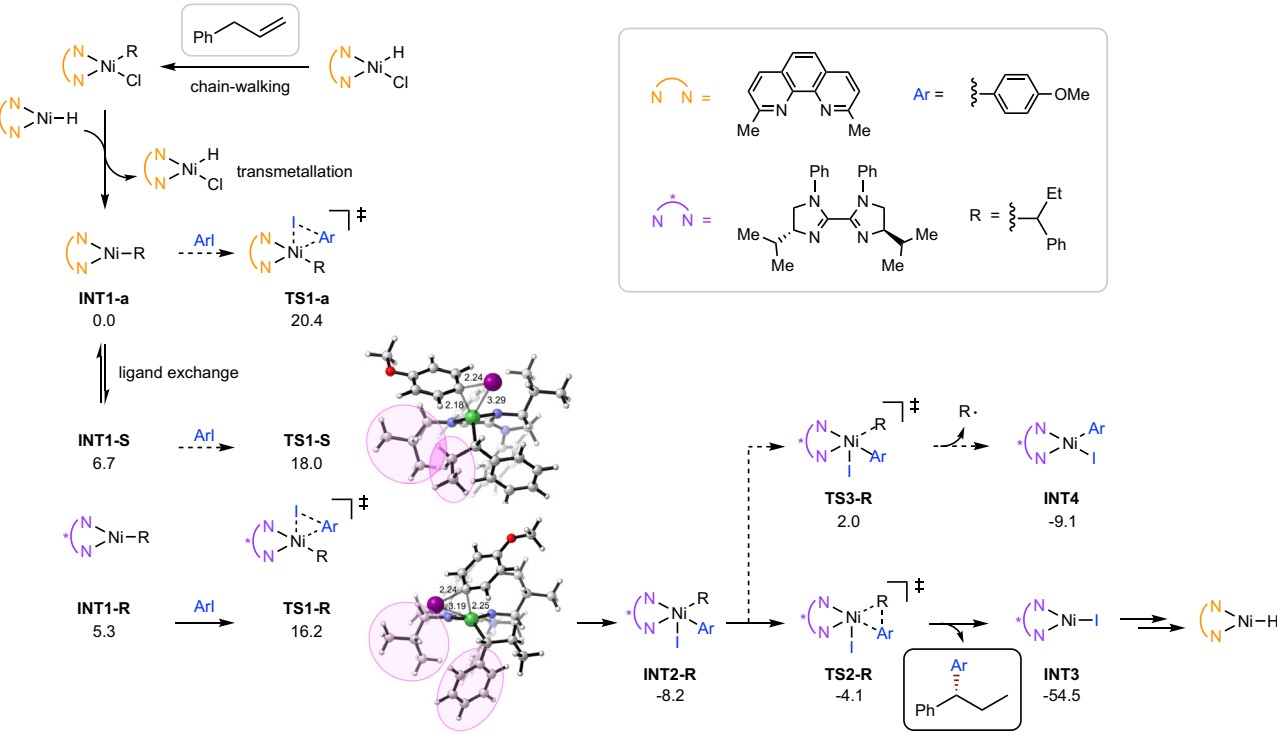

**Fig. 7 Computational study of the asymmetric arylation.** Computed at SMD(DCM)-(U)M06/6-311 + G(d,p)[SDD for Ni and I]//(U)B3LYP/6-31G(d) [LANL2DZ for Ni and I]. Values are relative Gibbs free energies in kcal/mol. Coloured atoms in graphics: grey, C; white, H; red, O; blue, N; green, Ni; purple, I. Distances are in angstroms.

## Data availability

The authors declare that the main data supporting the findings of this study, including experimental procedures and compound characterization, are available within the article and its supplementary information files, and also are available from the corresponding author. CCDC 2122000 contains the supplementary crystallographic data for **4t**. These data can be obtained free of charge from The Cambridge Crystallographic Data Centre via www.ccdc.cam.ac.uk/data_request/cif.

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

## Acknowledgements

Support was provided by NSFC (92156004, 21822105, 22001118), NSF of Jiangsu Province (BK20190281, BK20211555, BK20201245, BK20200300), the Fundamental Research Funds for the Central Universities (020514380253, 020514380263), programs for high-level entrepreneurial and innovative talents introduction of Jiangsu Province (group program), and Open Research Fund of School of Chemistry and Chemical Engineering, Henan Normal University. We thank the High Performance Computing Center (HPCC) of Nanjing University for doing the numerical calculations in this paper on its blade cluster system.

## Author contributions

S.Z. designed and supervised the project. Y.H., H.S., Y.Z. and Y.W. performed and analysed the experiments. J.M. and Y.L. designed, carried out and analysed the computational experiments. All authors co-wrote the manuscript, analysed the data, discussed the results, commented on the manuscript, and approved the final version of the manuscript.

## Competing interests

The authors declare the following competing interest(s): S.Z., Y.Z., and Y.H. are inventors on a patent application number CN202110888015.6 which is based on multiligand relay catalysis.
