## [Peer Review File · Nature Communications]

REVIEWER COMMENTS

Reviewer #1 (Remarks to the Author):

Zhu and co-workers described a novel approach for asymmetric hydroarylation of alkenes based on cascade chain-walking isomerization and iodobenzenes cross-coupling. The synergistic uses of two ligands to achieve successful regiocontrolled chain-walked isomerization and enantioselective C-C cross-coupling is particularly noteworthy. The procedure reported here seems to be operationally simple for typical synthetic laboratories without specialized reagents and chemicals. This procedure is likely to attract significant attention since regiocontrolled C-H bond functionalization remains a burgeoning field. Especially, enantiocontrolled C-C bond formation is highly challenging. The results presented here are truly outstanding. In my view, this work constitutes a breakthrough in this field.

Despite these fantastic results, I have a few queries for the authors to consider when revising this manuscript.

(1) The authors depict somewhat optimized results or finding at the beginning of the main text. While figure 2 shows some results derived from conditions that deviated from the standard, I found it difficult to appreciate the evolution of this work. Moreover, the Supplementary Material does not include detailed screening results as well. I suggest that the authors disclose more detailed screening results in the main text. A more elaborate discussion of these results should be worthwhile to guide readers to appreciate the underlying chemistry.

(2) Figure 2 shows that without L, L* can mediate effective 3a formation, but lower yield (52%) and lower regioselectivity. What is the other regioisomer? Would it be the proximal arylation product? What is the mass balance for this reaction without L?

(3) When L* was absent (i.e., only L was employed as the ligand), 3a was obtained in 4% yield. I wonder if the 4% product yield was derived from the condition with only 1 mol% L being used, or a total of 6 mol% L being used. Please clarify.

(4) The relative ratio of L and L* seems to be important for the enantioselectivity. As long as more L* is present, good %ee was observed. What is the mechanistic explanation for this finding? Noted that higher ligand concentration of L* would not automatically translate to higher Ni complexes concentration of L*, is it possible to verify the relative abundance of the Ni complexes under the optimized conditions?

(5) The use of chiral bis(imidazoles) ligands is very successful in this example. What are the results when the more classical bis(oxazolines) ligands were used. Moreover, apart from isopropyl as substituents, how the enantioselectivity would change in the hydroarylation reactions.

(6) Assuming Ni(L)-H be the hydride agents for hydroamidation, how the enantioselective arylation of the Ni-alkyl complex be possible? Presumably, some Ni(L*) complexes must be involved, and it should be some alkyl-Ni(L*) complexes. If the Ni(L) complex is particularly reactive for the chain-walk isomerization; clearly an exchange of L* ligand with L on the alkyl-Ni complex would be needed. What is the driving force of the ligand exchange process?

(7) Following point (6), what is the enantiodetermining step? hydride transfer step or the arylation step? For the arylation step, would it be the reductive elimination be the enantiodetermining step?

In summary, this manuscript presented excellent results that contribute significant to the Ni-catalyzed C-C bond cross coupling. I would recommend publication of this manuscript after major

revision.

Reviewer #2 (Remarks to the Author):

This work by Zhu et al. reported an enantioselective NiH-catalyzed remote hydroarylation of arene-containing alkenes with aryl halides. This protocol provides an efficient synthetic route to versatile enantioenriched 1,1-diarylalkanes. The key to the success of this asymmetric reaction is combination of two types of ligands. Although Zhu's group has recently demonstrated the same strategy in the construction of enantioenriched α -aryl alkylboronates (Chem 2021, 7, 3171–3188), this work has an indisputable novelty given its importance of construction of enantioenriched 1,1-diarylalkanes. Wide substrate scope and good functional group tolerance were observed. Several control experiments were also carried out to determine the plausible reaction pathway. The manuscript is well written and the presentation can easily be followed. The Experimental in the SI is detailed enough to allow reproduction, the analytics are adequate, and the NMR spectra reveal a satisfying product purity. Based on the above observations, this reviewer recommends to publish above results after minor revisions as follows.

- 1) Isotopic labeling experiments are essential for the reactions.
- 2) To confirm that achiral ligand promote chain walking and chiral ligand promote asymmetric arylation, this reviewer suggest that the authors carry out a control experiment like Chem 2021, 7, 3171–3188, Scheme 2B.
- 3) What happens if *o*-substituted aryl and heteroaryl iodides or bromides is present in Figure 3?
- 4) In the references section has some examples of incorrect formatting such as reference 32 'vinylarenes with aryl iodides. Angew. Chem. Int. Ed. 59, 21530–21534 (2020)' and reference 48 'J. Am. Chem. Soc., 141, 5628–5634 (2019).'
- 5) Some of the NMR spectra are not pristine clear. I believe there are some impurities in there. Examples: 3d, 3l, 3k and 4m.

Reviewer #3 (Remarks to the Author):

This paper describes the use of nickel catalysis to achieve the remote asymmetric hydroarylation of alkenes [containing a (hetero)aryl group] with aryl iodides to form enantioenriched 1,1-diarylalkenes. The key point is the use of two different ligands for nickel – one to promote NiH-catalyzed chain walking, and one to achieve asymmetric arylation. The scope of the process has been well-explored with respect to both substrates (alkene and aryl iodide), and in general, good yields and high enantioselectivities are obtained. Additional experiments involving regioconvergent arylation of isomeric mixtures of alkenes and application of the multiple ligand concept to 1,3-arylboration, are also described. Although all of the key concepts have been described before in the literature and the author has reported several similar nickel-catalyzed reactions, the specific combination described here to this particular reaction is novel and interesting. The study has been carried out well and the work is of high quality. I think it should appeal to a good number of readers and I am highly supportive of publication in Nature Communications.

There are only a few suggestions for improvement of the manuscript.

- (a) For the regioisomeric ratio "rr", the authors need to define what this means exactly. Does it mean the ratio of the major isomer vs the sum of the all the possible minor isomers?
- (b) Is there an explicit statement of how the authors proved the absolute configuration of the products? I know there is an X-ray structure of one of the products, but a clear statement would be beneficial to readres (apologies if I missed it).
- (c) In Figure 4, the absolute configuration of product 4q looks incorrect.
- (d) The SI is of good quality. For compound 3d, the abbreviation "NHTFA" would be better off

written as NHCOCF₃ as in the main paper itself. On the top of page 47, of the SI, there is a typographical error: "Non-3-en-1-ylbenzenethe"

In response to **reviewer 1** (quotes from reviewer are italicized):

Reviewer #1 (Remarks to the Author):

Zhu and co-workers described a novel approach for asymmetric hydroarylation of alkenes based on cascade chain-walking isomerization and iodobenzenes cross-coupling. The synergistic uses of two ligands to achieve successful regiocontrolled chain-walked isomerization and enantioselective C-C cross-coupling is particularly noteworthy. The procedure reported here seems to be operationally simple for typical synthetic laboratories without specialized reagents and chemicals. This procedure is likely to attract significant attention since regiocontrolled C-H bond functionalization remains a burgeoning field. Especially, enantiocontrolled C-C bond formation is highly challenging. The results presented here are truly outstanding. In my view, this work constitutes a breakthrough in this field.

Despite these fantastic results, I have a few queries for the authors to consider when revising this manuscript.

1. The authors depict somewhat optimized results or finding at the beginning of the main text. While figure 2 shows some results derived from conditions that deviated from the standard, I found it difficult to appreciate the evolution of this work. Moreover, the Supplementary Material does not include detailed screening results as well. I suggest that the authors disclose more detailed screening results in the main text. A more elaborate discussion of these results should be worthwhile to guide readers to appreciate the underlying chemistry.

We have now modified Figure 2 in the main text to be more informative, a detailed conditions screening and discussion results were added. As shown in Supplementary Tables 1–6 in Supplementary Information (page S45–S48), we have also added the more detailed conditions optimization results.

2. Figure 2 shows that without L, L can mediate effective 3a formation, but lower yield (52%) and lower regioselectivity. What is the other regioisomer? Would it be the proximal arylation product? What is the mass balance for this reaction without L?*

We have now added this information in Supplementary Information (page S48, Supplementary Table 6). Without L, chain-walking is slow and a large amount of the proximal arylation products could be observed (liner arylation product as the major). The mass balance for this reaction without L is also shown as below.

3. When L was absent (i.e., only L was employed as the ligand), 3a was obtained in 4% yield. I wonder if the 4% product yield was derived from the condition with only 1 mol% L being used, or a total of 6 mol% L being used. Please clarify.*

As shown in the modified Figure 2, we have now added this information. When only 1 mol% L was used, 4% yield was obtained (entry 3). When a total of 6 mol% L was used, 77% yield was obtained (entry 4).

4. The relative ratio of L and L seems to be important for the enantioselectivity. As long as more L* is present, good %ee was observed. What is the mechanistic explanation for this finding? Noted that higher ligand concentration of L* would not automatically translate to higher Ni complexes concentration of L*, is it possible to verify the relative abundance of the Ni complexes under the optimized conditions?*

Both the achiral ligand L and chiral ligand L* could promote the arylation process.

However, the arylation rate with chiral ligand L^* ligated is significantly faster than that of achiral ligand L ligated. We believe that the ligand exchange is in constant equilibrium, and higher concentration of L^* would lead to higher concentration of NiL^* complex. To gain insight into these details, more in-depth mechanistic studies are required. These studies are currently underway and details will report in due course.

5. The use of chiral bis(imidazoles) ligands is very successful in this example. What are the results when the more classical bis(oxazolines) ligands were used. Moreover, apart from isopropyl as substituents, how the enantioselectivity would change in the hydroarylation reactions.

As shown in the modified Figure 2, we have now added this information. When the more classical bis(oxazolines) ligands were used, a lower ee was obtained (80% ee, entry 9). When using isopropyl substituted bis(oxazolines) ligand $L3^*$, only moderate ee was obtained (55% ee, entry 10).

6. Assuming $Ni(L)-H$ be the hydride agents for hydroarylation, how the enantioselective arylation of the Ni -alkyl complex be possible? Presumably, some $Ni(L^)$ complexes must be involved, and it should be some $alkyl-Ni(L^*)$ complexes. If the $Ni(L)$ complex is particularly reactive for the chain-walk isomerization; clearly an exchange of L^* ligand with L on the $alkyl-Ni$ complex would be needed. What is the driving force of the ligand exchange process?*

We believe that the ligand exchange is in constant equilibrium, and the the arylation rate with chiral ligand L^* ligated is significantly faster than that of achiral ligand L ligated. As shown in Fig. 6c, the hydronickellation is not the enantiodetermining step. Supported by both preliminary experimental evidence and DFT calculation (Fig. 7), selective oxidative addition of one configuration of benzylic- $Ni(L^*)$ with $Ar-I$, followed by a stereo-specific reductive elimination would lead to single enantiomer of product.

7. Following point (6), what is the enantiodetermining step? hydride transfer step or the arylation step? For the arylation step, would it be the reductive elimination be the enantiodetermining step?

As mentioned above (point 6), currently we believe that the enantiodetermining step is the selective oxidative addition instead of reductive elimination. We have now carried out DFT calculation (Fig. 7), and this result further supports our hypothesis. To gain more direct experimental evidence, more in-depth mechanistic studies is still under investigation in our laboratory. Progress in this area will be reported in due course.

In response to **reviewer 2** (quotes from reviewer are italicized):

Reviewer #2 (Remarks to the Author):

This work by Zhu et al. reported an enantioselective NiH -catalyzed remote hydroarylation of arene-containing alkenes with aryl halides. This protocol provides an efficient synthetic route to versatile enantioenriched 1,1-diarylalkanes. The key to the success of this asymmetric reaction is combination of two types of ligands. Although Zhu's group has recently demonstrated the same strategy in the construction of enantioenriched α -aryl alkylboronates (Chem 2021, 7, 3171–3188), this work has an indisputable novelty given its importance of construction of enantioenriched 1,1-diarylalkanes. Wide substrate scope and good functional group tolerance were observed. Several control experiments were also carried out to determine the plausible reaction pathway. The manuscript is well written and the presentation can easily be followed. The Experimental in the SI is detailed enough to allow reproduction, the analytics are adequate, and the NMR spectra reveal a satisfying product purity.

Based on the above observations, this reviewer recommends to publish above results after minor revisions as follows.

1. Isotopic labeling experiments are essential for the reactions.

As shown in Fig. 6c, we have now carried out the isotopic labeling experiments. When styrene substrate 1b' was employed, a diastereomeric mixture of hydroarylation products were obtained with or without L. This indicates that the hydronickellation is not the enantiodetermining step.

2. To confirm that achiral ligand promote chain walking and chiral ligand promote asymmetric arylation, this reviewer suggest that the authors carry out a control experiment like Chem 2021, 7, 3171–3188, Scheme 2B.

As shown in Fig. 6b, we have now carried the suggested experiment. Without chain-walking ligand L, the alkene isomerization process is significantly slow.

3. What happens if o-substituted aryl and heteroaryl iodides or bromides is present in Figure 3?

When o-substituted aryl and heteroaryl iodides or bromides were used, low yields were obtained under current conditions. We have now added this information in the Supplementary Information (page S43).

4. In the references section has some examples of incorrect formatting such as reference 32 'vinylarenes with aryl iodides. Angew. Chem. Int. Ed. 59, 21530–21534 (2020)' and reference 48 'J. Am. Chem. Soc., 141, 5628–5634 (2019).'

We have now fixed several typos.

5. Some of the NMR spectra are not pristine clear. I believe there are some impurities in there. Examples: 3d, 3l, 3k and 4m.

We have now re-purified the compounds **3d**, **3l**, **3k** and **4m** and re-characterized the corresponding ¹H NMR and ¹³C NMR spectra.

In response to **reviewer 3** (quotes from reviewer are italicized):

Reviewer #3 (Remarks to the Author):

This paper describes the use of nickel catalysis to achieve the remote asymmetric hydroarylation of alkenes [containing a (hetero)aryl group] with aryl iodides to form enantioenriched 1,1-diaryllkenes. The key point is the use of two different ligands for nickel – one to promote NiH-catalyzed chain walking, and one to achieve asymmetric arylation. The scope of the process has been well-explored with respect to both substrates (alkene and aryl iodide), and in general, good yields and high enantioselectivities are obtained. Additional experiments involving regioconvergent arylation of isomeric mixtures of alkenes and application of the multiple ligand concept to 1,3-arylboration, are also described. Although all of the key concepts have been described before in the literature and the author has reported several similar nickel-catalyzed reactions, the specific combination described here to this particular reaction is novel and interesting. The study has been carried out well and the work is of high quality. I think it should appeal to a good number of readers and I am highly supportive of publication in Nature Communications.

There are only a few suggestions for improvement of the manuscript.

1. For the regioisomeric ratio “rr”, the authors need to define what this means exactly. Does it mean the ratio of the major isomer vs the sum of the all the possible minor isomers?

Yes, we have now added this definition “regioisomeric ratio (rr) represents the ratio of the major (1,1-diaryllkane) product to the sum of all other isomers as determined by GC and GC-MS analysis” in the legends of Fig. 2 and Fig. 3.

2. Is there an explicit statement of how the authors proved the absolute configuration of the

products? I know there is an X-ray structure of one of the products, but a clear statement would be beneficial to readers (apologies if I missed it).

As shown in the legend of Fig. 2. we have now added an explicit statement about the absolute configuration of the products “the absolute configuration was determined by chemical correlation or by analogy.”.

3. In Figure 4, the absolute configuration of product 4q looks incorrect.

The absolute configuration of **4q** has been revised accordingly.

4. The SI is of good quality. For compound 3d, the abbreviation “NHTFA” would be better off written as NHCOCF3 as in the main paper itself. On the top of page 47, of the SI, there is a typographical error: “Non-3-en-1-ylbenzenethe”

We have carefully double-checked the SI and corrected all the typos mentioned above.

Shaolin Zhu

REVIEWERS' COMMENTS

Reviewer #1 (Remarks to the Author):

This manuscript by Liang, Wang and Zhu describes the enantioselective Ni-catalyzed migratory alkene hydroarylation enabled by synergistic uses of a chain-walking ligand and a chiral arylation ligand. This work is novel, and it is a well-conducted investigation, especially much more data have been incorporated in the main text and the Supporting Information. To make a convincing case, the authors included DFT calculation and found that the oxidative addition step should be enantio-determining. This part of the study was not included in the earlier version of the manuscript.

Further to the authors' mechanistic claim, I have the following remark:

(1) The NiH(L) is fast for alkene hydrometallation and chain-walking isomerization. As indicated in the DFT calculation, this process should afford (R)- and (S)-alkylNi(L) complexes. Assuming the authors are correct, L-L* exchange should produce a diastereomeric mixture of (R)- and (S)-alkylNi(L*) complexes. According to the authors' DFT calculation, only the diastereomer of (R)-alkylNi(L*) undergo facile oxidative addition by aryl iodide and react further down to products.

My remark is that there be a missing step that allows the diastereomeric (R)- and (S)-alkylNi(L*) to interconvert. Otherwise, the reaction should produce at most 50% yield based on initial alkenes plus 50% of the isomerized alkenes. This is not consistent with the authors' findings. Would the authors comment on this?

(2) The authors' DFT calculation showed that the alkylNi(L) is less reactive than alkylNi(L*) toward oxidative addition to aryl iodide. Would the authors comment on the origin of the reactivity difference between the alkylNi complexes on the two ligands?

In conclusion, I support publication of this work in Nature Communications after minor revision.

Reviewer #2 (Remarks to the Author):

In the revised manuscript, because the comments raised by the reviewer have been properly answered, the current manuscript may be accepted with only the following corrections.

1) Line 284, Nature 563, 379-383 (2018). The formatting of the hyphen(-) is inconsistent with that used in other references.

Reviewer #3 (Remarks to the Author):

In this revised submission, the authors have satisfactorily addressed most of the reviewer comments, providing new discussion and/or experimentation to support this additional material. These include isotopic labeling experiments as well as DFT experiments, which provide useful mechanistic insight. Minor errors have also been corrected. There are only a few trivial points to address:

- For the response to Reviewer 2, point 3, there should be a statement in the main paper that o-substituted (hetero)aryl halides give lower yields.
- For the response to Reviewer 3, point 1, one notices that in the legend of Figure 3, there is repetition within the phrase "determined by GC and GC-MS analysis. were determined by GC and GC-MS analysis,". Please correct this text.
- For the response to Reviewer 3, point 2, the added text seems insufficient. The authors have mentioned "the absolute configuration was determined by chemical correlation or by analogy", but correlation or analogy to what? The authors need to say that the absolute configuration of 4t was determined by X-ray

crystallography, and the configurations of the remaining products were assigned by analogy. Or some suitable alternative text.

Other than that, publication of this fine paper in Nature Communications is recommended.

In response to **reviewer 1** (quotes from reviewer are italicized):

Reviewer #1 (Remarks to the Author):

This manuscript by Liang, Wang and Zhu describes the enantioselective Ni-catalyzed migratory alkene hydroarylation enabled by synergistic uses of a chain-walking ligand and a chiral arylation ligand. This work is novel, and it is a well-conducted investigation, especially much more data have been incorporated in the main text and the Supporting Information. To make a convincing case, the authors included DFT calculation and found that the oxidative addition step should be enantio-determining. This part of the study was not included in the earlier version of the manuscript.

Further to the authors' mechanistic claim, I have the following remark:

(1) The NiH(L) is fast for alkene hydrometallation and chain-walking isomerization. As indicated in the DFT calculation, this process should afford (R)- and (S)-alkylNi(L) complexes. Assuming the authors are correct, L-L exchange should produce a diastereomeric mixture of (R)- and (S)-alkylNi(L*) complexes. According to the authors' DFT calculation, only the diastereomer of (R)-alkylNi(L*) undergo facile oxidative addition by aryl iodide and react further down to products. My remark is that there be a missing step that allows the diastereomeric (R)- and (S)-alkylNi(L*) to interconvert. Otherwise, the reaction should produce at most 50% yield based on initial alkenes plus 50% of the isomerized alkenes. This is not consistent with the authors' findings. Would the authors comment on this?*

We sincerely thank the reviewer for his/her comment. The unreacted **INT1-S** [(S)-alkylNi(L*)] can be rapidly converted to **INT1-R** [(R)-alkylNi(L*)] via a reversible ligand exchange, transmetallation and alkene insertion process (see Supplementary Fig. 5, also shown below). We have added a sentence in the revised main text and the details in the revised SI.

Supplementary Fig 5. Conversion of **INT1-S** to **INT1-R** via a reversible ligand exchange, transmetalation and alkene insertion process. Computed at SMD(DCM)-(U)M06/6-311+G(d,p)[SDD for Ni and I] //(U)B3LYP/6-31G(d)[LANL2DZ for Ni and I]. Values are relative Gibbs free energies in kcal/mol. When X = Cl (from NiCl₂), the interconversion needs an activation free energy of 18.2 kcal/mol, which is very close to that of the undesired oxidative addition via **TS1-S** (18.0 kcal/mol). However, as the reaction proceeds, LNiI generates, which can undergo the exchange reaction with LNiHCl to give LNiHI. This process is approximately neutral in energy (+0.5 kcal/mol). When X = I, the barrier for the interconversion is lowered to 13.2 kcal/mol, ensuring a fast conversion of **INT1-S** to **INT1-R**.

(2) *The authors' DFT calculation showed that the alkylNi(L) is less reactive than alkylNi(L*) toward oxidative addition to aryl iodide. Would the authors comment on the origin of the reactivity difference between the alkylNi complexes on the two ligands?*

Compared with bipyridine ligand (**L**), the bisimidazoline ligand (**L***) is a better electron donor, thus facilitating the Ni(I)/Ni(III) oxidative addition of alkylnickel(I) with aryl iodide.

In conclusion, I support publication of this work in Nature Communications after minor revision.

In response to **reviewer 2** (quotes from reviewer are italicized):

Reviewer #2 (Remarks to the Author):

In the revised manuscript, because the comments raised by the reviewer have been properly answered, the current manuscript may be accepted with only the following corrections.

1) Line 284, Nature 563, 379-383 (2018). The formatting of the hyphen(-) is inconsistent with that used in other references.

We have now fixed this typo.

In response to **reviewer 3** (quotes from reviewer are italicized):

Reviewer #3 (Remarks to the Author):

In this revised submission, the authors have satisfactorily addressed most of the reviewer comments, providing new discussion and/or experimentation to support this additional material. These include isotopic labeling experiments as well as DFT experiments, which provide useful mechanistic insight. Minor errors have also been corrected. There are only a few trivial points to address:

- For the response to Reviewer 2, point 3, there should be a statement in the main paper that o-

substituted (hetero)aryl halides give lower yields.

We certainly appreciate the comment made by the reviewer. As suggested, we have mentioned the aryl halide profile in the text “However, o-substituted (hetero)aryl halides gave lower yields under current conditions.”

- For the response to Reviewer 3, point 1, one notices that in the legend of Figure 3, there is repetition within the phrase “determined by GC and GC-MS analysis. were determined by GC and GC-MS analysis,”. Please correct this text.

The reviewer is absolutely right, and we have now deleted the repetition accordingly.

- For the response to Reviewer 3, point 2, the added text seems insufficient. The authors have mentioned “the absolute configuration was determined by chemical correlation or by analogy”, but correlation or analogy to what? The authors need to say that the absolute configuration of 4t was determined by X-ray crystallography, and the configurations of the remaining products were assigned by analogy. Or some suitable alternative text.

We sincerely thank the reviewer for his/her comment. As suggested, we have now added an explicit statement about the absolute configuration of the products “the absolute configuration of 4t was determined by X-ray crystallography, and the configurations of the remaining products were assigned by analogy.”.

Other than that, publication of this fine paper in Nature Communications is recommended.

Shaolin Zhu